# The Microbiome in PDAC—Vantage Point for Future Therapies?

**DOI:** 10.3390/cancers14235974

**Published:** 2022-12-02

**Authors:** Nina Pfisterer, Catharina Lingens, Cathleen Heuer, Linh Dang, Albrecht Neesse, Christoph Ammer-Herrmenau

**Affiliations:** 1Department of Gastroenterology, Gastrointestinal Oncology and Endocrinology, University Medical Center Goettingen, 37075 Goettingen, Germany; 2Clinical Research Unit KFO5002, University Medical Center Goettingen, 37075 Goettingen, Germany; 3Department of Medical Bioinformatics, University Medical Center Goettingen, 37075 Goettingen, Germany

**Keywords:** microbiome, pancreatic ductal adenocarcinoma, contamination, biomarker

## Abstract

**Simple Summary:**

Pancreatic cancer is a highly lethal cancer and less than 10% of patients survive the 5-year mark. The molecular and biological underpinnings leading to this dismal prognosis are well-described, however, translation of these findings with subsequent improvement of the poor prognosis has been slow. The complex and dynamic accumulation of microbes, also called the microbiome, has recently attracted scientific interest in the pathogenesis of several diseases including pancreatic cancer. Since then, a limited number of significant findings were published pointing towards an important role of the microbiome in cancer, in particular pancreatic cancer. Here, we provide a concise synopsis of the current findings focusing exclusively on pancreatic cancer, and also highlight the pitfalls of microbiome research for scientists as well as clinicians to foster standardization and comparability amongst microbiome studies.

**Abstract:**

Microorganisms have been increasingly implicated in the pathogenesis of malignant diseases, potentially affecting different hallmarks of cancer. Despite the fact that we have recently gained tremendous insight into the existence and interaction of the microbiome with neoplastic cells, we are only beginning to understand and exploit this knowledge for the treatment of human malignancies. Pancreatic ductal adenocarcinoma (PDAC) is an aggressive solid tumor with limited therapeutic options and a poor long-term survival. Recent data have revealed fascinating insights into the role of the tumoral microbiome in PDAC, with profound implications for survival and potentially therapeutic outcomes. In this review, we outline the current scientific knowledge about the clinical and translational role of the microbiome in PDAC. We describe the microbial compositions in healthy and tumoral pancreatic tissue and point out four major aspects of the microbiome in PDAC: pathogenesis, diagnosis, treatment, and prognosis. However, caution must be drawn to inherent pitfalls in analyzing the intratumoral microbiome. Among others, contamination with environmental microbes is one of the major challenges. To this end, we discuss different decontamination approaches that are crucial for clinicians and scientists alike to foster applicability and physiological relevance in this translational field. Without a definition of an exact and reproducible intratumoral microbial composition, the exploitation of the microbiome as a diagnostic or therapeutic tool remains theoretical.

## 1. Introduction

Traditionally, cancer has been appreciated as a genetic disease where mutations or deletions of oncogenes and tumor suppressor genes result in the transformation of cells and uncontrolled growth. However, besides genetic alterations, numerous additional factors, such as epigenetic modifications or adaptations of the tumor immune system, can contribute to the development and progression of human tumors. Substantial research progress has been made during the last couple of decades, resulting in a profound understanding of the underlying molecular mechanisms of tumor evolution and therapeutic resistance. Furthermore, the improved understanding of the molecular underpinnings has resulted in surveillance efforts (e.g., for breast and colon cancer) and novel therapeutic options that have considerably improved the overall prognosis of cancer patients during the last decade. However, despite intense research efforts, the mortality of PDAC is still one of the highest for all solid tumors, showing a 5-year survival rate of below 10% [1]. A European study has even predicted it to become the third leading cause of cancer-related deaths by 2025 [2]. Clinically, this can be attributed to the late diagnosis, early reoccurrence, and high metastatic rate of PDAC that is accompanied by a high-intrinsic resistance to systemic therapies and radiotherapy [3]. Indeed, given that a resection is the only curative treatment option, the tumor recurs in most cases within two years, leading to a median survival of 24–30 months [4]. However, most patients (>80%) are diagnosed with metastasized or locally advanced stages of the disease [5]. In the palliative setting, the treatment options are limited. The establishment of FOLFIRINOX in metastatic PDC was a significant improvement, pushing the median survival to 11.1 months compared with the 6.8 months achieved through gemcitabine alone [6]. For patients with a poor performance status, gemcitabine can be administered in combination with nab-paclitaxel, which results in a median survival of 8.6 months [7].

To tackle this devasting prognosis, intensive research is being conducted to understand the tumor biology. PDAC is characterized by genetic heterogeneity and a complex tumor microenvironment (TME) [8]. The key players of the TME are cancer-associated fibroblasts, immune cells, extracellular matrix components such as collagen and hyaluronic acid, tumor vessels, and nerves that make up the majority of the tumor bulk [9]. Therapeutic resistance against most conventional chemotherapeutics has often been attributed to these phenotypic properties; however, therapeutic depletion strategies of TME components such as hyaluronan, collagen, or targeting of pro-fibrotic signaling pathways (e.g., sonic hedgehog) has not yielded significant clinical benefits [3]. Most recently, a new player in this highly dynamic network system has been introduced and was even added as a new hallmark of cancer: the microbiome [10].

The microbiome subsumes a community of microorganisms (bacteria, fungi, viruses, and archaea) occupying a habitat and their structural elements such as nucleic acids and metabolites [11]. Current calculations estimate the absolute number of microbial cells to equal the eukaryotic cells in humans, numbering ~3 × 10^3^. To this end, it is well-known that the large intestines harbor the majority of microbes by far [12]. In healthy individuals, the microbiome exerts a plethora of physiological functions contributing to the homeostasis of the organism [13,14]. However, probably the most significant finding of recent years by several groups was that tumors also harbor their own specific microbiome, which differs amongst tumor entities and is distinct to healthy tissue [15,16,17]. Hence, the microbiome has emerged as a novel component of interest for basic and translational science, and intense efforts are undertaken to better understand its role during tumorigenesis and progression and to exploit the microbiome as a prognostic and therapeutic target.

The most basic example of how microbes may be involved in oncogenesis is the long-known principle of chronic infection potentially leading to cancer formation. In fact, it was reported that pathogens are directly responsible for over 16% of all cancer cases [18]. For some infectious agents, the mechanisms are well-described: e.g., epithelial injury or chronic inflammation leading to genetic alterations [19]. However, it is increasingly being recognized that not only are microbes acquired via infection but the host’s commensal microbiome is also capable of having an impact on carcinogenesis. 

The advent of the next-generation sequencing (NGS) technique has dramatically improved the scope of microbiome research, outcompeting conventional culturing methods. At the same time, however, the high sensitivity of the NGS method may also bear the risk of detecting environmental contaminations that significantly confound the findings. Especially in low biomass samples such as tumors, contaminants easily dominate the biological signal of species truly present in the sample [20,21,22]. This requires very stringent controls for the contamination introduced during the entire study process from sample acquisition to bioinformatic evaluation. Nevertheless, despite the growing amount of data, open questions remain. These questions include whether the various host site microbiomes, e.g., the intestinal microbiome, may serve as a biomarker for cancer, whether the tumoral microbiome causes or promotes cancer, or whether it is regulated and exploited by the tumor itself. A major focus lies on the modulation of the tumor immune system by microorganisms and subsequent mechanisms of therapy resistance [16,23]. 

Here, we attempt to review the current knowledge of this highly relevant and rapidly evolving field, including the functional and therapeutical aspects of the microbiome in PDAC. We aim to provide a state-of-the-art reference for scientists and clinicians who are interested in tumoral microbiome research including the current definition of the intratumoral microbiome and methodological pitfalls. In particular, the existence of environmental bacteria raises concerns about the true microbial composition of the tumor. Thus, this review will also accurately describe thorough decontamination approaches.

## 2. The Microbiome in Healthy and Tumoral Pancreatic Tissue

For some organs such as the skin, small and large intestines, and oral cavity, bacterial colonization is a common feature [24]. However, most internal organs were long thought to be sterile. It is now broadly accepted that organs linked to the gut system harbor a low biomass of microbiota [16,25]. However, it is unknown whether microbial sites (upper or lower gastrointestinal tract) are associated with the pancreatic microbiome. The first studies started to investigate how microbes reach the pancreas. By fluorescently administering labeled bacteria or fungi to mice via the oral gavage, it was shown that microbes are capable of migrating to the pancreas via the duodenum within a couple of hours [16,26].

### 2.1. The Normal Pancreatic Microbiome

The discovery of the existence of a tumor microbiome and the fact that it is significantly different from non-malignant tissue was groundbreaking [16,17]. However, publications describing the physiological pancreatic microbiome are very rare. Most studies use tissue adjacent to the tumor or pancreatic samples from benign diseases as controls. Geller et al. first detected bacterial 16S rRNA in healthy pancreas samples from organ donors by a quantitative real-time polymerase chain reaction (qRT-PCR) analysis; however, they only found bacterial DNA in 15% of these control samples as opposed to 76% in PDAC samples [23]. Attempts to identify microbiota specific to the healthy pancreatic microbiome were made by Pushalkar et al. in 2018. They found *Chlamydiales* and *Brevibacterium* in normal human pancreatic tissue in an increased relative abundance as compared with PDAC material [16]. Contrary to this, Thomas et al. did not find any significant differences between healthy pancreatic and tumor tissue, although the authors observed higher species abundances of *Acinetobacter*, *Enterobacter*, and *Pseudomonas* in the normal pancreas [27]. Eventually, del Castillo et al. reported a higher mean relative abundance of *Lactobacillus* in a human non-cancerous pancreas [17].

### 2.2. The Tumoral Pancreatic Microbiome

The pancreatic tumor microbiome has been investigated in multiple studies. Table 1 provides an overview on the pancreatic microbiota described by recent studies. Regarding the microbial signature in PDAC, most studies came up with a highly similar microbiota composition, especially on a higher taxonomic rank. The dominant taxon across most published microbial compositions in PDAC is the phylum *Proteobacteria*. This comprises *Gammaproteobacteria* including *Enterobacteriaceae*, commensal bacteria inhabiting the intestines, and *Alphaproteobacteria*. A plethora of environmental genera belong to the latter class: *Rhizobium*, a symbiont of legumes; *Sphingopyxis*, found in environmental niches such as water and soil; and *Methylobacterium*, a common contaminant of DNA extraction kit reagents [20,28]. Surprisingly, these three genera and many more environmental bacteria were often published as tumoral microbes (Table 1). This observation raises a major concern and pitfall of sequencing low microbial biomass samples: contamination. Contaminants are introduced during every step of the microbiome analysis workflow and potentially mask the true taxa in tumoral samples if not properly controlled for (Figure 1). Therefore, the RIDE checklist (Report–Include–Determine–Explore), a minimum standards checklist for low microbial biomass microbiome studies proposed by Eisenhofer et al., may guide all future microbiome studies. It stipulates to report methodology, include controls, determine the level of contamination, and explore the impact of contamination in downstream analyses [21]. The implication of negative controls is the most essential step for controlling for the contamination introduced in every part of the study process.

### 2.3. Decontamination Is Indispensable for Low Microbial Biomass Samples

In terms of contamination control, Nejman et al. published a remarkable study presenting a thoroughly decontaminated data set [15]. The authors implemented more than 800 negative controls and 6 filters to properly clean up over 1500 low biomass samples. First, highly prevalent species that were present in controls were removed (filter 1). A per condition filter for every single process step was applied, which removed contaminants from DNA extraction (filter 2), PCR batch (filter 3), sequencing lane batch (filter 4), as well as paraffin contaminants (filter 5). The last filter removed center-specific contaminants. With this approach, the initial species count (9190) was cut down to 528 bacterial species distributed over 7 different cancer types. Indeed, for pancreatic cancer, less than 10 species were predominantly identified per sample, and only 25 different genera were determined in 67 PDAC samples. Another study by Chakladar et al. exclusively focused on the pancreatic microbiome and its associations with carcinogenesis and prognosis [32]. The authors retrieved the NGS RNA sequencing data of 187 PDAC patients from The Cancer Genome Atlas (TCGA). Moreover, a very different decontamination approach as compared with Nejman et al. was conducted due to the lack of negative controls. Chakladar et al. applied three filters and referred to two publications that reported lab- and hospital-born contaminants, respectively [34,35]. Based on these lists, the authors entirely removed the phyla *Firmicutes* and *Bacteroidetes* and the genus *Fusobacterium*. This approach is questionable, as it is not evident if the samples were subject to the reagents analyzed by Glassing et al. or derived from the centers assessed by Rampelotto et al. [34,35]. Furthermore, with the exclusion of *Firmicutes* and *Bacteroidetes*, the authors excluded the two most abundant orointestinal phyla [24]. According to Nejman et al., these taxa account for 25% of tumoral bacteria [15]. Next, microbes with little abundance variation among different sequencing depths were removed. In other words, microbes were considered contaminants whose abundance did not increase with increasing read number. This approach underlies the following assumption: contaminants do not correlate with total read numbers but affect all samples equally. This assumption defines the basis of the frequency method in the decontam package, an easy to use and renowned R tool. However, Davis et al. noted that this particular approach is not suitable for low biomass microbiome studies in which contaminants are equally as abundant or even more abundant than true species [22]. Finally, microbes with a noticeably high abundance at specific sequencing dates were removed. After applying all three of these filters, more than 200 taxa were declared contaminants and subsequently removed from the data set [32]. Generally, the key message is that thorough decontamination of microbial sequencing data is indispensable. However, the particular method of decontamination is still under debate. We strongly recommend, in line with the RIDE checklist, the introduction of negative controls at every sample processing step. 

## 3. The Role of the Microbiome in Pancreatic Carcinogenesis

In 2012, the International Agency for Research on Cancer (IARC) Working Group on the Evaluation of Carcinogenic Risks to Humans reported that about 13% of all global cancer cases are caused by so-called “oncomicrobes”. Eleven distinctly defined microbes evidently induce cancer, and there is experimental evidence for even more [36]. Contrary to these well-defined oncomicrobes in certain tumor entities, there is emerging evidence that the tumoral microbiome contributes to carcinogenesis in different ways. Figure 2 illustrates the established and putative associations between the microbiota and oncogenesis.

### 3.1. Microbes Activate Oncogenic Signaling

Generally, intense research efforts have been undertaken on contact-dependent and contact-independent interactions of microorganisms impacting tumorigenesis in the last couple of years. This includes the presentation and secretion of virulence factors, signaling induced via physical binding to host cells, and the creation of a pro-inflammatory TME via immune cell recruitment [37]. In gastric cancer, for instance, the contact-dependent mechanism of *Helicobacter pylori* leading to the neoplastic transformation of epithelial cells is well-established and has already been described several decades ago by the Correa pathway [38]. *H. pylori* interferes with the Wnt/β-catenin pathway, ultimately affecting cellular turnover and apoptosis [37]. Silva-García et al. have reviewed Wnt/β-catenin signaling dysregulation by popular pathogens, of which some were also found in PDAC (Table 1) such as *Pseudomonas*, *Clostridium*, *Bacteroides*, *Escherichia*, *Haemophilus*, and *Shigella*. The authors concluded that the secretion of virulence factors from pathogenic bacteria may lead to alterations in cell proliferation, apoptosis, and inflammation-associated cancer via various molecular signaling strategies that have also been extensively reviewed elsewhere [37,39]. Another intriguing study by Kadosh et al. found that mutant p53 had tumor-suppressive functions in the proximal gut but oncogenic effects in the distal gut. Mechanistically, it was described that the disruption of the Wnt pathway led to tumor-suppressive effects of mutant p53 via the prevention of chromatin binding by transcription factor 4. The tumor-suppressive properties were entirely eradicated by a single gut microbiome-derived metabolite, gallic acid. These findings emphasize that the microbiome is capable of regulating the functional outcome of cellular mutations [40].

### 3.2. Direct Carcinogenic Effects of Microbes via Mutagenesis

Meanwhile, microbiota may drive carcinogenesis via direct genetic alterations. They influence genomic stability and thereby contribute to shaping the cancer genome [41,42]. Geng et al. performed in vitro experiments in oral squamous cell carcinoma cells that they infected with *Fusobacterium nucleatum* and detected DNA double-strand breaks (DSB) as indicated by the expression of γH2AX. Because DSB increases the likelihood of tumor onset and development, the authors concluded that *F. nucleatum* promoted cell proliferation by DNA damage via the Ku70/p53 pathway [41]. Tumor suppressor *p53* is frequently mutated in PDAC. In particular, *p53* arginine mutations were found in pancreatic cancer patients at a high rate [43]. Notably, arginine mutations were also found in *Kras*, the earliest and most frequently mutated oncogene in PDAC. It was speculated that peptidylarginine deiminases derived from oral bacteria which cause periodontitis, such as *Porphyromonas gingivalis*, cause *p53* and *Kras* point mutations via arginine degradation [44,45,46]. Because cancers mostly arise from somatic mutations and because there is emerging evidence for microbiota substantially contributing to these mutational mechanisms, microbe-driven mutagenesis is a highly relevant research field that will undergo significant advancements in understanding the origins of the cancer genome in the near future [42]. 

### 3.3. Indirect Impact of Microbes via Chronic Inflammation

There are theories that microbes have an indirect impact on tumorigenesis. These are mainly accompanied by chronic inflammation in the oral cavity [47,48,49]. These studies report an association of the oral microbiome and periodontitis with the development of pancreatic cancer. However, all of them, which were cited several times before, suffer from significant pitfalls. Michaud et al. and Farrell et al. use low sensitivity methods, such as antibodies or microarrays/qRT-PCR to determine the oral microbiome [48,49]. Moreover, the association study by Michaud et al. lacks information regarding alcohol consumption, an important confounding variable which might explain the link between periodontitis and pancreatic cancer [47]. All three studies also do not provide information about the incidence of chronic pancreatitis.

Nevertheless, there is evidence that chronic inflammation might contribute to tumorigenesis. Ochi et al. suggested that inflammatory microbe-associated molecular patterns (MAMP), such as lipopolysaccharide (LPS), are involved in the promotion of pancreatic cancer [50]. Evolutionary conserved MAMPs may derive from commensals or pathogens and can be systemically recognized by the innate immune system via the so-called pattern recognition receptors (PRR), of which, for example, toll-like receptor (TLR) 4 is the PRR of LPS [51,52]. Further evidence that LPS enhances carcinogenesis via the promotion of epithelial-to-mesenchymal transition and angiogenesis was provided by other groups [53,54,55]. However, the exact mechanisms by which the microbiome-derived MAMPs promote PDAC remain elusive at present, and the link between periodontitis and PDAC has to be further proven [37].

### 3.4. Microbiome–Immune System–Axis Involvement in Oncogenesis

The immunogenic TME reprogramming at the tumor site is a potential mechanism of how microbes contribute to carcinogenesis given the predominantly inflammatory TME in PDAC [16,56]. However, it is still under debate whether the present microorganisms act in a tumor-promoting or tumor-repressing mode on the immune system. Most likely, both directions are featured depending on the microbial composition in the respective setting. Using a PDAC mouse model, Pushalkar et al. showed that the depletion of the microbiome enabled the modulation of immune cell composition, including the reduction of myeloid-derived suppressor cells and increased activation of Th1-type CD4^+^ and cytotoxic CD8^+^ T cells, leading to a significantly reduced tumor burden in mice [16]. In accordance with Pushalkar et al., Chakladar et al. associated high microbial abundance in PDAC patients with immunosuppression including low M2 macrophages and T cells, activation of oncogenic pathways, and downregulation of tumor-suppressive pathways. The authors found 13 microbes that were associated with dysregulated gene signatures including oncogenic methylation, cancer progression, and immune system modulation [32].

Apart from bacteria, it was also published that the mycobiome, the fungal part of the microbiome, has an impact on pancreatic oncogenesis. Aykut et al. reported that fungi derived from the gut play a role in PDAC pathogenesis and reported a 3000-fold increase of fungi in pancreatic tumors compared with healthy pancreatic tissue [26]. Most interestingly, the PDAC mycobiome was significantly different from the gut or normal pancreas mycobiome composition. Comparably to Pushalkar et al. who demonstrated a significant reduction of the tumor burden by performing bacterial ablation, Aykut et al. also reported the protective properties against tumor growth of mycobiome ablation. Eventually, the authors discovered the mechanism of pathogenic fungi activating mannose-binding lectin to be the driving mechanism for the complement cascade and thereby tumor promotion [26]. Most recently, a connecting line between the fungal microbiome and immunogenic aspects in PDAC pathogenesis was drawn by Alam et al.; in short, they found mycobiome-enhanced interleukin (IL)-33 secretion by cancer cells, which attracted and activated T helper 2 and innate lymphoid 2 cells in the TME. The genetic deletion of IL-33 or antifungal treatment reduced T helper cells, resulting in tumor regression [57]. 

### 3.5. Influence of Microbial Metabolites on Oncogenesis

Finally, microbes are capable of influencing oncogenesis via their metabolites. This contact-independent mechanism of remote production and secretion of bioactive molecules into the systemic circulation can influence tumors and metastasis at distant sites. For example, short-chain fatty acids (SCFAs), derived from bacterial fermentation, may potentially alter the microbial composition in the gut via pH regulation in addition to immune modulation [58,59,60,61]. It was intensively reviewed that SCFA-producing bacteria may act on epigenetics, gene expression, cell proliferation, and apoptosis in colonic cancer [62]. Due to the often-reported involvement of microbial metabolites in carcinogenesis and anticancer potential, their diagnostic potential is to be examined in the near future and may hold promising results.

## 4. Diagnostic Aspects of the Microbiome in PDAC

### 4.1. Difficulties in Establishing Screening Tools for PDAC

Considering the available descriptive and preliminary mechanistic findings on the PDAC tumor microbiome, the question of its potential diagnostic value and possible implication as a biomarker may arise. One of the main problems with PDAC is most often the late-stage diagnosis as the tumor is often locally advanced or metastasized. This is mostly due to a lack of early-stage symptoms. To date, a reliable screening method for pancreatic cancer is not available in the clinical routine [63]. Studies investigating different site-specific microbiomes, such as the oral and fecal microbiome, point towards a possible application of the microbiome as a diagnostic biomarker in PDAC [49,64]. 

### 4.2. The Orointestinal Microbiome as PDAC Biomarker

Indeed, there are numerous publications addressing the microbiome in the oral cavity and its diagnostic potential for PDAC, of which the latest are summarized in Table 2. One of the largest studies was published by Fan et al., which was a population-based nested case-control study on the predictive power of the oral microbiome to assess the risk for pancreatic cancer [65]. Over 730 oral wash samples from two prospective cohort studies were evaluated. The authors found oral pathogens such as *Porphyromonas gingivalis* to be associated with an increased pancreatic cancer risk. The pitfall of the microbial patterns of the oral cavity, however, is their rather pronounced heterogeneity and low specificity, as they may also be present in other cancer entities [66]. Microbiome studies present contradictory results concerning the microbial composition and differential abundances of these microbes (Table 2). This can be mainly ascribed to the different kinds of sampling methods, e.g., sputum, dorsal tongue, buccal, or gingival swabs. Furthermore, due to different sequencing approaches, i.e., depending on the selected variable (V) region of the 16S rRNA gene, the results significantly vary [67]. 

Many studies have demonstrated a positive correlation between the pancreas and gut microbiome. For example, Ren et al. reported that the gut microbiome analyzed via stool samples was unique in PDAC and may serve as a non-invasive biomarker for the diagnosis of this disease [70]. Recently, Kartal et al. explored the fecal and salivary microbiota in PDAC patient samples from a Spanish and German case-control study as potential biomarkers; they found 27 fecal species that could be employed to identify PDAC throughout early and late stages with high accuracy. Thus, the authors suggested the fecal microbiome as a feasible early-stage PDAC biomarker, particularly in combination with carbohydrate antigen 19-9 [64]. However, these findings require validation in larger patient cohorts. Only a few months later, Nagata et al. reused the data from Kartal et al. and added their Japanese cohort dataset, which also included oral and gut bacteriophages [84]. Their aim was to further identity oral and gut metagenomic microbial signatures to predict PDAC. The authors found 30 gut and 18 oral species to be significantly associated with PDAC in their newly introduced Japanese cohort, and their metagenomic classifiers were also able to predict PDAC accurately. Consistently with Kartal et al., Nagata et al. found the gut microbiomes of European and Asian patients to present a globally robust and powerful biomarker for identifying PDAC. 

Taken together, the orointestinal microbiome might be used as a non-invasive screening tool. However, the translational implication to the clinical setting remains unclear at present. Given the high cost of sequencing, a multiplex PCR or microarray for those identified bacteria might be more feasible. Furthermore, it must be discussed who will be screened, whether it be only high-risk patients or a broader screening population. Further studies with high sample numbers, such as one already completed in the U.S. (NCT03302637), will hopefully provide answers to these questions.

### 4.3. Blood-Derived Microbial Signatures as PDAC Biomarker

One of the most common sampling techniques in the clinical routine is blood drawing. Bacterial extracellular vesicles (bEV) are nano-sized, lipid membrane-delimited particles that contain different molecules, such as DNA, metabolites, proteins, and lipids. Recently, there is growing evidence that bEVs play an important role in bacteria–bacteria and bacteria–host communication [85,86]. These bEVs can be detected in the host’s blood, urine, bile, and stool. The exploitation of these vesicles for therapeutic and diagnostic purposes is still in its infancy [87]. One recent study revealed the diagnostic value of bEVs for differentiating between benign and malignant tumors [88]. Another Korean study performed a retrospective propensity score matching analysis showing a distinguishable composition of bEVs in blood by 16S rRNA sequencing [89]. Here again, environmental bacteria were detected in peripheral blood, emphasizing the need for thorough decontamination protocols for blood samples as well, in cases where bacterial DNA is found in very low concentrations [85]. Poore et al. demonstrated that microbial plasma profiles in over 10,000 patients, which were different from their respective healthy tissue signature, can predict different cancer types [90]. The authors used whole-genome and whole-transcriptome sequencing studies from TCGA. Moreover, pre-diagnosis blood samples from PDAC patients were subject to oral microbiota antibody measurements in a study by Michaud et al. Indeed, high levels of antibodies against *Porphyromonas gingivalis*, the pathogen responsible for periodontitis, was correlated with a two-fold increased PDAC risk [48].

## 5. Influence of the Microbiome on Treatment and Prognosis of PDAC

Microbiome profiling could indeed be a feasible way for performing prognostic assessments in clinics. Various studies have already evaluated the prognostic power of the (tumoral) microbiome in PDAC [30,32,33,91]. Moreover, altering the microbiome for therapy may pave the way for new therapeutic approaches.

### 5.1. Microbial Impact on Anti-Tumor Therapies

It has repeatedly been published that the microbiome plays a remarkable role in response to chemotherapy, immunotherapy, and even radiotherapy [23,92,93,94,95]. Regarding immunotherapy, immune checkpoint inhibitors (ICIs), e.g., programmed cell death protein 1 (PD-1), its ligand PD-L1 or cytotoxic T lymphocyte-associated protein 4 (CTLA-4) antibodies, are employed to interrupt the inhibitory effects of tumor cells towards T cells [96,97]. In 2018, Routy et al. reported that an abnormal intestinal microbial composition contributed to primary ICI resistance in cancer patients. Moreover, the clinical response to ICIs correlated with the relative abundance of *Akkermansia muciniphila* in patient stool samples. The enhancement of the anti-tumoral PD-1 blockage effect in mice could be achieved via fecal microbiome transplantation (FMT) from ICI responders but not via FMT from non-responders [95]. Recently, phase 1 clinical trials were published that investigated the safety and feasibility of FMT in anti-PD-1-refractory metastatic melanoma. Here, three out of ten patients and six out of fifteen patients showed a clinical response, respectively [98,99]. However, only less than 1% of PDACs are susceptible to ICIs, and a number of ongoing phase 1 and 2 trials are testing different combinations of ICIs and conventional chemotherapies [100,101]. Whether microbiome modulation via FMT can change ICI responsiveness needs to be further investigated. 

A growing body of evidence for microbial influence on chemotherapy efficiency was published in the last decade, which we have summarized in Table 3. A landmark study by Geller et al. found *Mycoplasma hyorhinis* derived from human dermal fibroblasts cocultured with colorectal cancer (CRC) and PDAC cell lines to confer gemcitabine resistance. This was also confirmed in vivo when the transplantation of *M. hyorhinis*-infected CRC cells to mice flanks rendered them gemcitabine-resistant. The authors further showed that the long isoform of cytidine deaminase expressed by *Gammaproteobacteria* was responsible for inactivating gemcitabine, causing tumors harboring the respective microbes to become more chemoresistant [23]. This finding is highly relevant clinically as gemcitabine, alone or in combination with nab-paclitaxel, is the only treatment option in the palliative settings of patients with poor performance scores. Lehouritis et al. reported the in vitro and in vivo modulation of anticancer drugs by *Escherichia coli* and *Listeria welshimeri*. Depending on the drug, either an enhancement or reduction in drug efficacy was found [102]. Iida et al. found oxaliplatin to be less effective in germ-free/antibiotics-treated mice [103]. Oxaliplatin is part of FOLFIRINOX, the most potent chemotherapy regimen for treating PDAC compared with gemcitabine monotherapy [6]. Iida et al. found that in the absence of microbes, pro-inflammatory genes were downregulated after oxaliplatin treatment. Therefore, the authors concluded that inflammation, which was correlated with the fecal microbiota composition, promoted the anti-tumoral effect of oxaliplatin. To this end, depending on the occurring species and chosen chemotherapy, the drug response was enhanced or inhibited. However, the translation of these study results to clinical application has yet to be shown.

Another study addressing this issue by Guo et al. associated microbial communities with either the classical or basal-like subtype, respectively, of which it is published that the former has a better prognosis and response to FOLFIRINOX [33,116]. Thereby, Guo et al. proved the predictive value of the microbiome via the subtype-dependent microbiome compositions. However, it is important to mention that to this day, these transcriptome-based subtypes have not been implemented into the clinics and do not (yet) play a role in patient stratification towards personalized therapy [117].

### 5.2. Microbiome Modulation Approaches

There are several conceivable ways to exploit the microbiota for cancer treatment. Considering the previously described findings on the differences between the healthy and tumor microbiomes, the most straightforward idea is to modulate the microbiome. To this end, unfavorable microbial signatures should be ablated, and beneficial communities could be enhanced. Microbiome modulation in order to improve anticancer therapy may be achieved via FMT, pre-/pro-/post- and antibiotics, dietary modifications, and phage therapy approaches in order to enhance anticancer therapy [23,95,118,119,120,121]. The latter represents a potent option for targeting specific bacteria with minimum off-target effects; this is an almost impossible challenge with antibiotic treatment, which usually eradicates a broad spectrum of microbiota. In addition to this selective microbiome-altering effect, bacteriophages can be equipped with chemotherapy nanoparticles to be directly released in the TME in CRC-bearing mice and improve the efficiency of chemotherapy [121]. PDAC may benefit from this technique as chemotherapy delivery to the tumor cells can be impeded due to the pronounced stroma and poor vascularization of the tumor mass [122]. Tanoue et al. defined a specific group of 11 bacterial strains from healthy human donor feces capable of inducing non-inflammatory anticancer immunity, mostly via the induction of interferon-γ-producing CD8^+^ T cells. Murine syngenic tumor models were colonized with this bacterial composition which enhanced ICI efficacy, and the authors concluded that there was biotherapeutic potential for this gut microbiota composition [119]. A very recent study by Panebianco et al. tested a postbiotic (bacterial metabolite) of intestinal bacteria, namely butyrate, which is already known for its anticancer and anti-inflammatory properties in PDAC cell lines and mouse models [114]. The authors showed that butyrate enhanced gemcitabine effectiveness by decreasing the proliferation and induction of apoptosis in vitro and by reducing cancer-associated stroma modulation, a hallmark of PDAC in vivo. Therefore, the authors concluded that the supplementation with such postbiotics would ameliorate treatment efficacy. Kesh et al. most recently found that FMT from PDAC-implanted control mice to PDAC-implanted obese mice rendered the tumors of the latter more sensitive towards chemotherapy [115]. S-adenosyl methionine-producing bacteria were enriched in the feces of mice who received the control diet, whereas queuosine (Q)-producing bacteria were elevated in the high fat diet-fed mice. Interestingly, the treatment of pancreatic cancer cells with Q increased PRDX1, which is protective against oxidative stress induced by chemotherapy. This emphasizes why dietary modifications may be beneficial for enhancing anticancer therapy. 

In a recently published hallmark study, Riquelme et al. analyzed the intratumoral microbiome of PDAC patients with long-term survival (LTS) compared with short-term survivors (STS). Remarkably, the authors identified a specific microbial signature with long-term PDAC survival, indicating the potential of the microbiome to predict the patient outcome [30]. Moreover, Riquelme et al. performed FMT experiments from human healthy controls, STS and LTS to previously antibiotics-treated mice, and subsequently transplanted these mice with PDAC tumor cells. Feces from the LTS significantly reduced the tumor volume as compared with STS feces, suggesting that the microbiome of LTS is causally responsible for tumor growth and subsequent survival.

### 5.3. The Microbiome as Potential Biomarker for PDAC Prognosis

The lack of valid therapeutic biomarkers was addressed by Guenther et al., who evaluated LPS as a surrogate marker for gemcitabine efficacy in PDAC [91]. Following the Geller study on bacteria-mediated chemotherapy resistance, Guenther et al. found LPS, a surrogate for bacterial colonization, to negatively predict adjuvant gemcitabine efficacy [23,91]. In addition to chemotherapy resistance, early metastasis to the liver is a major clinical problem in PDAC. Chakladar et al. found intratumoral bacteria, mainly *Proteobacteria*, to be associated with metastasis and a poorer prognosis. Particularly, *Acinetobacter baumanni* and *Mycoplasma hyopneumoniae* were overrepresented in smokers. Because the authors used RNA sequencing data, it was possible to associate these species with different pathways. Both bacteria were correlated with an upregulation of oncogenic and downregulation of tumor-suppressive pathways [32]. Interestingly, a REVEALER plot shows significant correlation of *M. hypopneumoniae* and deletions of tumor-suppressive gene loci. However, it must be emphasized that the microbial composition presented by Chakladar et al. requires further validation. Most of the specifically mentioned species are common environmental bacteria or, in case of *M. hypopneumoniae*, a well-known porcine enzootic pneumonia pathogen [123].

### 5.4. Perspectives of the Microbiome–PDAC Axis

Several studies are registered at ClinicalTrials.gov, which will further elucidate the role of the microbiome in PDAC. Most of them evaluate the microbiome from different body sites as a potential biomarker. Four trials (NCT04274972, NCT04922515, NCT05523154, and NCT04931069) are collecting oral and rectal samples for anticipating surgical complications prior to pancreatic surgery. Another two studies (NCT05580887, NCT04922515) are investigating the intestinal microbiome as predictors of response to common chemotherapies in PDAC. Only one clinical trial (NCT05462496) has implemented a microbiome modulation protocol with pembrolizumab and/or antibiotics after neoadjuvant chemotherapy and before resection. Hereby, the investigators aim to determine the intratumoral immune response following different microbiome-altering approaches.

## 6. Conclusions

It was only during the last couple of years that a new player was introduced to the oncology research field: the microbiome. Consistently, several studies have revealed that PDAC harbors its own tumoral microbiome, which impinges on carcinogenesis, response to chemotherapy, and prognosis. The interactions between the microbiome and the tumor immune system are emerging and may offer new vantage points for therapeutic interventions. However, regarding the specific microbial composition, the majority of studies are inconsistent. Thus, a thorough decontamination protocol is highly mandatory for any microbiome studies. An exact definition of the tumor microbial composition is especially relevant with respect to targeted therapeutic interventions. To this end, sophisticated approaches to specifically alter the tumor-influencing microbiome, such as bacteriophages, fecal microbiome transplantations, or the substitution of certain strains, are still highly theoretical. Until now, the microbiome is not ready to be exploited as a diagnostic or therapeutic tool. Recently registered clinical trials focus on the potential role of the microbiome as a biomarker for prognosis and surgical outcome, further elucidating the role of the microbiome in PDAC.

## Figures and Tables

**Figure 1 cancers-14-05974-f001:**
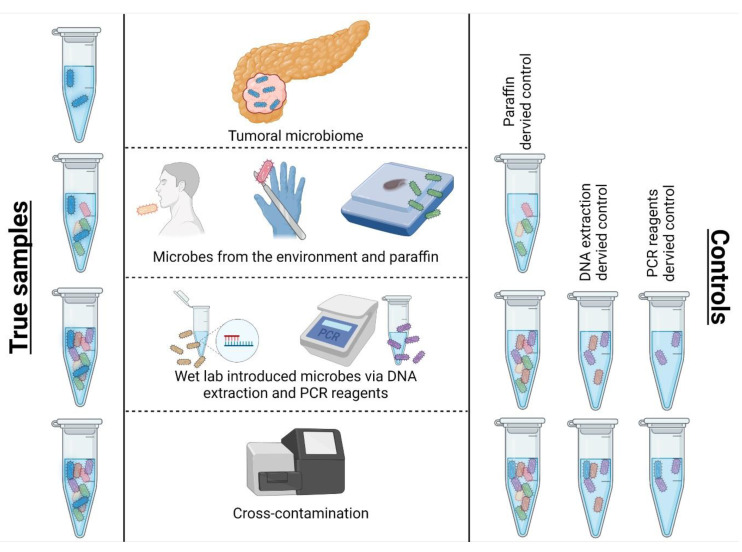
Sources of contamination and recommended controls in tumoral microbiome analysis. Microbial contamination during sample preparation is impossible to avoid. Considering the low biomass of tumor microbiomes, contaminants from the environment, paraffin (in case of FFPE samples), reagents from DNA extraction kits, PCR, and library preparation can outnumber the real tumor-derived microbial count. Thus, a thorough decontamination protocol should be mandatory. Ideally, negative controls are implemented at each processing step for every single sample batch. However, real tumoral taxa can also occur in negative controls due to computational cross-contamination. Here, an index switch in multiplexed analysis is introduced by sequencing errors (barcode leakage). Desoxyribonucleic acid (DNA), polymerase chain reaction (PCR), formalin-fixed paraffin-embedded (FFPE).

**Figure 2 cancers-14-05974-f002:**
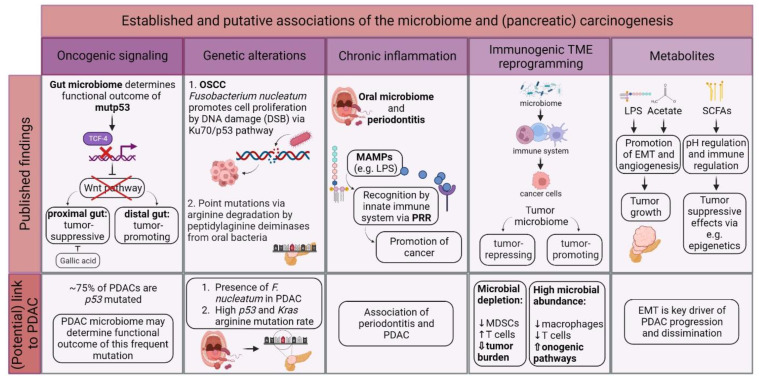
Potential involvement of the microbiome in (pancreatic) oncogenesis. There is growing evidence on how different microbiomes contribute to carcinogenesis, e.g., via promoting oncogenic signaling, direct and indirect genetic alterations, chronic inflammation, and interaction with the immune system and secretion of microbe-derived metabolites. However, most of these theories have yet to be validated in PDAC patients. Tumor microenvironment (TME); mutant p53 (mutp53); pancreatic ductal adenocarcinoma (PDAC); oral squamous cell carcinoma (OSCC); desoxyribonucleic acid (DNA); double-strand break (DSB); microbe-associated molecular pattern (MAMP); lipopolysaccharide (LPS); pattern recognition receptor (PRR); myeloid-derived suppressor cell (MDSC); short-chain fatty acid (SCFA); epithelial-to-mesenchymal transition (EMT); and pondus hydrogenii (pH).

**Table 1 cancers-14-05974-t001:** Overview on pancreatic microbiota (including fungi) described in recent studies.

Year	Authors	Tissue	Microbiota	Sequencing Method	Ref.
2017	Geller et al.	Human PDAC tumors (fresh samples)	*Enterobacteriaceae* (f) *Pseudomonadaceae* (f) *Moraxellaceae* (f) *Streptococcaceae* (f) *Enterococcaceae* (f) *Staphylococcaceae* (f) *Carnobacteriaceae* (f) *Corynebacteriaceae* (f) *Micrococcaceae* (f) *Microbacteriaceae* (f)	16S rRNA amplicon sequencing (V4 region) (NGS)	[23]
2017	Li et al.	Human pancreatic cyst fluid (fresh frozen samples)	*Fusobacterium* (g) *Ruminococcus* (g) *Staphylococcus* (g) *Caldimonas* (g) *Arthrobacter* (g) *Acinetobacter* (g) *Bacteroides* (g) *Orpinomyces* (g) *Anaerococcus* (g) *Escherichia/Shigella* (g) *Acidaminococcus* (g) *Coprococcus* (g) *Collinsella* (g) *Butyricicoccus* (g) *Parabacteroides* (g) *Alistipes* (g) *Clostridium XI* (g) *Gemmiger* (g) *Dorea* (g) *Lachnospiracea incertae sedis* (g) *Blautia* (g) *Bifidobacterium* (g) *Sphingomonas* (g) *Faecalibacterium* (g)	Sanger sequencing of PCR products of universal 16S rRNA primers, 16S rRNA amplicon sequencing (V3–V4 variable region) (NGS)	[29]
2018	Pushalkar et al.	Human PDAC tumors (fresh frozen samples)	*Proteobacteria* (p) (*Pseudomonas, Elisabethkingia*) *Bacterioidetes* (p) *Firmicutes* (p) *Actinobacteria* (p/c)	16S rRNA amplicon sequencing (V3–V4 hyper-variable region) (NGS)	[16]
		Human normal pancreas (fresh frozen samples)	*Chlamydiales* (o) *Brevibacterium* (g)		
2019	Aykut et al.	Human PDAC tumors (fresh samples)	*Ascomycota* (p) *Basidiomycota* (p) (*Malassezia*)	18S rRNA amplicon sequencing (ITS1 region) (NGS)	[26]
		Murine KC pancreatic tumors	*Ascomycota* (p) *(Aspergillus, Cladosporium, Penicillium, Stenocarpella*, *Alternaria*, *Mycosphaerella*, *Fusarium*, *Ascochyta*, *Xeromyces*, *Saccharomycopsis*, *Stagonosporopsis*) *Basidiomycota* (p) (*Ustilago*, *Naganishia*, *Tilletia*, *Vishniacozyma*, *Sporobolomyces*, *Tritirachium*, *Malassezia*) *Mucor* (g)		
2019	Riquelme et al.	Human PDAC tumors (FFPE samples)	*Gammaproteobacteria* (c) *Bacilli* (c) *Actinobacteria* (p/c) *Clostridia* (c) *Bacteroidia* (c) *Alphaproteobacteria* (c) *Betaproteobacteria* (c) *Shingobacteria* (c) *Negativicutes* (c) *Flavobacteria* (c) *Erysipelotrichia* (c) *Cytophagia* (c) *Coriobacteria* (c) *Fusobacteria* (c) *Verrucomicrobiae* (p) *Deltaproteobacteria* (c) *Pseudoxanthomonas* (g) *Streptomyces* (g) *Saccharopolyspora* (g) *Bacillus clausii* (s)	16S rRNA amplicon sequencing (V4 region) (NGS)	[30]
2019	del Castillo et al.	Human pancreatic cancer (fresh frozen samples)	*Fusobacterium* (g) *Porphyromonas* (g) *Prevotella* (g) *Capnocytophaga* (g) *Selenomonas* (g)	16S rRNA amplicon sequencing (V3–V4 hyper-variable region) (NGS)	[17]
		Human non-cancer pancreas (fresh frozen samples)	*Lactobacillus* (g)		
2019	Gaiser et al.	Human pancreatic cyst fluid (fresh frozen samples)	*Gemella* (g) *Methylobacterium* (g) *Pasteurellaceae* (f) *Escherichia*/*Shigella* (g) *Propionibacteria* (g) *Bergeyella* (g) *Acinetobacter* (g) *Haemophilius* (g) *Eikenella* (g) *Lactobacillus* (g) *Enterococcus* (g) *Streptococcus* (g) *Granulicatella* (g) *Enterobacter* (g) *Enterobacteriaceae* (f) *Klebsiella* (g) *Prevotella* (g) *Staphylococcus* (g) *Cutibacterium* (g)	PacBio SMRT full-length 16S rRNA gene sequencing	[31]
2020	Nejman et al.	Human pancreatic cancer (fresh frozen and FFPE samples)	*Enterbacter asburiae* (s) *Klebsiella pneumoniae* (s) *Citerobacter freundii* (s) *Fusobacterium nucleatum* (s) *Enterbacter cloacae* (s)	16S rRNA amplicon sequencing (5 amplicons, including V2, V3, V5, V6, and V8, covering 68% of the gene) (NGS)	[15]
2020	Chakladar et al.	Human PDAC tumors	*Actinobacteria* (p/c) *Mycoplasma* (g) *Proteobacteria* (p) *(Shigella, Salmonella, Acinetobacter, Escherichia)*	NGS RNA sequencing data from TCGA	[32]
2021	Guo et al.	Human PDAC tumors (fresh frozen samples)	*Pseudomonas* (g) *Elizabethkingia* (g) *Acinetobacter* (g) *Brevundimonas* (g) *Sphingopyxis* (g) *Comamonas* (g) *Sphingomonas* (g) *Sphingobium* (g) *Caulobacter* (g) *Delftia* (g) *Agrobacterium* (g) *Klebsiella* (g) *Rhizobium* (g) *Bradyrhizobium* (g) *Dechloromonas* (g)	Whole-genome sequencing (NGS)	[33]

Phylum (p), class (c), order (o), family (f), genus (g), species (s), pancreatic ductal adenocarcinoma (PDAC), Svedberg unit (S), ribosomal ribonucleic acid (rRNA), variable (V), next-generation sequencing (NGS), polymerase chain reaction (PCR), Kras^G12D^;Cre^p48^ (KC), internal transcribed spacer between 18S and 5.8S rRNA (ITS1), formalin-fixed paraffin-embedded (FFPE), single-molecule real-time (SMRT), The Cancer Genome Atlas (TCGA).

**Table 2 cancers-14-05974-t002:** Summary of studies regarding the oral, intestinal, and fecal microbiome of patients as a non-invasive biomarker for pancreatic cancer.

Year	Authors	Study Design; Country of Conduction	Sample Type	Detection Method	Number of Patients	Change in Bacterial Composition	Ref.
2012	Farrell et al.	Prospective study; USA	Saliva	Microarray, qRT-PCR	38 PC 27 CP 38 HC	*Neisseria elongata, Streptococcus mitis* increased in PC cases	[49]
2013	Lin et al.	Cross-sectional study; USA	Oral wash samples	16S rRNA amplicon sequencing (NGS)	13 PC 3 CP 12 HC	*Bacteroides* increased in PC cases as compared with HC; *Corynebacterium, Aggregatibacter* decreased in PC cases as compared with HC	[68]
2013	Michaud et al.	Prospective study; European countries	Blood	Immunoblot array	405 PC 416 HC	Plasma IgG against *Porphyromonas gingivalis* ATCC 53978 increased in PC cases	[48]
2015	Torres et al.	Cross-sectional study; USA	Saliva	16S rRNA amplicon sequencing (V4 region) (NGS); qRT-PCR	8 PC 78 other diseases (including pancreatic disease, non-pancreatic digestive disease/cancer, and non-digestive disease/cancer) 22 HC	*Leptotrichia:Porphyromonas* ratio increased in PC cases; *Neisseria, Aggregatibacter* decreased in PC cases	[69]
2016	Fan et al.	Case-control study; USA	Oral wash samples	16S rRNA amplicon sequencing (V3–V4 region) (NGS)	361 PDAC 371 HC	*Porphyromonas gingivalis, Aggregatibacter actinomycetemcomitans* increased in PDAC cases; *Leptotrichia* decreased in PDAC cases	[65]
2017	Ren et al.	Prospective study; China	Feces	16S rRNA amplicon sequencing (V3–V5 region) (NGS)	85 PC 57 HC	*Veillonella, Klebsiella, Selenomonas*, LPS-producing bacteria (*Prevotella, Hallella, Enterobacter, Cronobacter*) increased in PC cases; *Bifidobacterium*, butyrate-producing bacteria (*Coprococcus, Clostridium IV, Blautia, Flavonifractor, Anaerostipes bifidum, Butyricicoccus, Dorea, Gemmiger*) decreased in PC cases	[70]
2017	Olson et al.	Cross-sectional study; USA	Saliva	16S rRNA amplicon sequencing (V4–V5 region) (NGS)	40 PDAC 39 IPMN 58 HC	*Firmicutes* (e.g., *Streptococcus*) increased in PDAC cases; *Proteobacteria* (e.g., *Haemophilus, Neisseria*) decreased in PDAC cases as compared with HC	[71]
2018	Pushalkar et al.	Case-control study; USA	Rectal swabs	16S rRNA amplicon sequencing (V3–V4 region) (NGS)	32 PDAC 31 HC	*Proteobacteria, Actinobacteria, Fusobacteria, Verrucomicrobia, Synergistetes, Euryarchaeota* increased in PDAC cases	[16]
2018	Mei et al.	Case-control study; China	Duodenal mucosa	16S rRNA amplicon sequencing (V3–V4 region) (NGS)	14 PC (pancreatic head cancer) 14 HC	*Acinetobacter, Aquabacterium, Oceanobacillus, Rahnella, Massilia, Delftia, Deinococcus, Sphingobium* increased in PC cases; *Porphyromonas, Paenibacillus, Enhydrobacter, Escherichia, Shigella, Pseudomonas* decreased in PC cases	[72]
2019	Lu et al.	Case- control study; China	Tongue coat samples	16S rRNA amplicon sequencing (V3–V4 region) (NGS)	30 PC (pancreatic head cancer) 25 HC	*Leptotrichia, Fusobacterium, Rothia, Actinomyces, Corynebacterium, Atopobium, Peptostreptococcus, Catonella, Oribacterium, Filifactor, Campylobacter, Moraxella, Tannerella* increased in PC cases; *Haemophilus, Porphyromonas, Paraprevotella* decreased in PC cases	[73]
2019	del Castillo et al.	Cross-sectional study; USA	Tissue samples (pancreatic duct, duodenum, pancreas); swabs (bile duct, jejunum, stomach); feces	16S rRNA amplicon sequencing (V3–V4 region) (NGS)	39 PC 12 periampullary cancer 18 non-cancer pancreatic conditions 8 non-cancer gastrointestinal conditions 34 HC	*Porphyromonas, Prevotella, Selenomonas, Gemella, Fusobacterium spp.* increased in cancer cases as compared with non-cancer cases; *Lactobacillus* decreased in cancer cases as compared with non-cancer cases	[17]
2019	Half et al.	Case-control study; Israel	Feces	16S rRNA amplicon sequencing (V3–V4 region) (NGS)	30 PDAC 6 pre-cancerous lesions 16 NAFLD 13 HC	*Veillonellaceae, Akkermansia, Odoribacter* increased in PDAC cases as compared with HC; *Clostridiacea, Erysipelotrichaeceae, Ruminococcaceae, Lachnospiraceae, Anaerostipes* decreased in PDAC cases as compared with HC	[74]
2020	Vogtmann et al.	Case-control study; Iran	Saliva	16S rRNA amplicon sequencing (V4 region) (NGS)	273 PDAC 285 HC	*Enterobacteriaceae, Lachnospiraceae* G7, *Bacteroidaceae, Staphylococcaceae* increased in PDAC cases; *Haemophilus* decreased in PDAC cases	[75]
2020	Sun et al.	Case-control study; China	Saliva	16S rRNA amplicon sequencing (V3–V4 region) (NGS)	10 PC 17 BPD 10 HC	*Fusobacteria* (e.g., *Fusobacterium periodonticum*)*, Bacteroidetes, Firmicutes* increased in PC cases; *Proteobacteria* (e.g., *Neisseria mucosa*) decreased in PC cases	[76]
2020	Kohi et al.	Case-control study; USA	Duodenal fluid	16S and 18S rRNA amplicon sequencing (16S V3–V4 rRNA region, 18S ITS1 rRNA region) (NGS)	74 PDAC 98 pancreatic cysts 134 HC	*Fusobacterium, Bifidobacterium* genera, *Enterococcus* increased in PDAC cases as compared with HC; *Escherichia/Shigella, Enterococcus, Clostridium sensu stricto 1, Bifidobacterium* increased in PDAC cases as compared with pancreatic cysts; *Fusobacterium, Rothia, Neisseria* increased in PDAC cases with short-term survival	[77]
2020	Wei et al.	Case- control study; China	Saliva	16S rRNA amplicon sequencing (V3–V4 region) (NGS)	41 PDAC 69 HC	*Leptotrichia, Actinomyces, Lachnospiraceae, Micrococcaceae, Solobacterium, Coriobacteriaceae, Moraxellaceae, Streptococcus, Rothia, Peptostreptococcus, Oribacterium* increased in PDAC cases; *Porphyromonas gingivalis, Fusobacteriaceae, Campylobacter, Spirochaetaceae*, *Veillonella, Neisseria, Selenomona, Tannerella forsythia, Prevotella intermedia* decreased in PDAC cases	[78]
2021	Zhou et al.	Case-control study; China	Feces	Metagenomic shotgun sequencing (NGS)	32 PDAC 32 AIP 32 HC	*Gammaproteobacteria* (e.g., *Escherichia coli*)*, Veillonella* (*V. atypica, V. parvula, V. dispar*)*, Clostridium* (e.g., *Clostridium bolteae, Clostridium symbiosum*)*, Fusobacterium nucleatum, Streptococcus parasanguinis, Prevotella stercorea* increased in PDAC cases as compared with HC; Butyrate-producing bacteria (*Eubacterium rectale, Faecalibacterium prausnitzii, Roseburia intestinalis, Coprococcus*), *Ruminococcus, Dialister succinatiphilus* decreased in PDAC cases as compared with HC	[79]
2021	Matsukawa et al.	Case-control study; Japan	Feces	Whole-genome sequencing (including PCR) (NGS)	24 PC (thereof 22 PDAC) 18 HC	*Klebsiella pneumoniae, Clostridium bolteae, Clostridium symbiosum, Streptococcus mutans, Alistipes shahii, Bacteroides, Parabacteroides, Lactobacillus* increased in PC cases	[80]
2021	Sugimoto et al.	Case-control study; Japan	Duodenal fluid	16S rRNA terminal restriction fragment length polymorphism method (5’ FAM-labeled 516F and 1510R primers)	22 benign pancreaticobiliary diseases (thereof 16 BPD) 12 pancreaticobiliary cancer (thereof 9 PC)	*Bifidobacterium, Clostridium* cluster XVIII increased in PC cases as compared with BPD	[81]
2022	Petrick et al.	Prospective study; USA	Oral wash samples	Metagenomic shotgun sequencing (NGS)	148 PDAC (thereof 122 of African Americans, 26 of Caucasians) 441 HC (thereof 354 of African Americans, 87 of Caucasians)	No significant changes in PDAC cases among African Americans; *Porphyromonas gingivalis* increased in PDAC cases among Caucasians; *Porphyromonas gingivalis, Prevotella intermedia, Tannerella forsythia* increased in PDAC cases among never-smokers	[82]
2022	Kartal et al.	Case- control study; Spain, Germany	Saliva	Metagenomic shotgun sequencing (NGS) (43 PDAC, 12 CP, 45 HC) 16S rRNA amplicon sequencing (V4 region) (NGS) (59 PDAC, 28 CP, 55 HC)	59 PDAC 28 CP 55 HC (Spanish cohort only)	No significant changes in PDAC cases	[64]
			Feces	Metagenomic shotgun sequencing (NGS) (101 PDAC, 29 CP, 82 HC) 16S rRNA amplicon sequencing (V4 region) (NGS) (51 PDAC, 23 CP, 46 HC)	101 PDAC 29 CP 82 HC (thereof 57 PDAC, 29 CP and 50 HC from Spanish cohort; 44 PDAC and 32 HC from German cohort)	*Streptococcus, Akkermansia, Veillonella atypica, Fusobacterium nucleatum/hwasookii, Alloscardovia omnicolens* increased in PDAC cases as compared with HC; *Romboutsia timonensis, Faecalibacterium prausnitzii, Bacteroides coprocola, Bifidobacterium bifidum* decreased in PDAC cases as compared with HC	
2022	Guo et al.	Case-control study; China	Feces	16S rRNA amplicon sequencing (27F, 1492R primer) (NGS)	36 resectable PDAC 36 unresectable PDAC	*Pseudonocardia, Cloacibacterium, Mucispirillum, Anaerotruncus* increased in unresectable PDAC cases; *Alistipes, Anaerostipes, Faecalibacterium, Parvimonas* decreased in unresectable PDAC cases	[83]
2022	Nagata et al.	Case-control study; Japan, Spain, Germany	Saliva	Metagenomic shotgun sequencing (NGS)	90 PDAC 280 HC (thereof 47 PDAC and 235 HC from Japanese cohort; others from Kartal et al., 2022)	*Firmicutes* (unknown *Firmicutes, Dialister* and *Solobacterium spp.*)*, Prevotella spp.* (*Prevotella pallens, Prevotella sp.* C561) increased in PDAC cases among Japanese cohort; *Streptococcus* spp. (e.g., *Streptococcus salivarius, Streptococcus thermophilus, Streptococcus australis*) decreased in PDAC cases among Japanese cohort; No significant changes in PDAC cases among Spanish cohort; No correlation for oral species between the Japanese and Spanish datasets	[84]
			Feces	Metagenomic shotgun sequencing (NGS)	144 PDAC 65 CP 150 IPMN 317 HC (thereof 43 PDAC, 65 CP, 150 IPMN and 235 HC from Japanese cohort; others from Kartal et al., 2022)	*Streptococcus oralis, Streptococcus vestibularis, Streptococcus anginosus, Veillonella atypica, Veillonella parvula, Actinomyces spp., Clostridium symbiosum,* unknown *Mogibacterium, Clostridium clostridioforme* increased in PDAC cases as compared with HC among Japanese cohort; Unknown *Lachnospiraceae, Eubacterium ventriosum,* unknown *Butyricicoccus, Faecalibacterium prausnitzii* decreased in PDAC cases as compared with HC among Japanese cohort; *Clostridium symbiosum, Streptococcus oralis*, unknown *Mogibacterium* increased in PDAC cases as compared with IPMN and CP among Japanese cohort; Significant correlation for gut species between the Japanese and Spanish datasets and between the Japanese and German datasets; *Streptococcus* spp. (*S. anginosus* and *S. oralis*), *Veillonella spp.* (*V. parvula* and *V. atypica*) increased in PDAC cases among all three cohorts; *Faecalibacterium prausnitzii* decreased in PDAC cases among all three cohorts	

United States of America (USA), quantitative real-time polymerase chain reaction (qRT-PCR), pancreatic cancer (PC), chronic pancreatitis (CP), healthy control (HC), Svedberg unit (S), ribosomal ribonucleic acid (rRNA), next-generation sequencing (NGS), immunoglobulin G (IgG), American Type Culture Collection (ATCC), variable (V), pancreatic ductal adenocarcinoma (PDAC), lipopolysaccharide (LPS), intraductal papillary mucinous neoplasm (IPMN), non-alcoholic fatty liver disease (NAFLD), benign pancreatic disease (BPD), internal transcribed spacer between 18S and 5.8S rRNA (ITS1), autoimmune pancreatitis (AIP), polymerase chain reaction (PCR), fluorescein amidite (FAM), forward (F), reverse (R).

**Table 3 cancers-14-05974-t003:** Summary of studies on bacteria, bacterial metabolites, or antibiotics influencing the response to chemotherapy.

Year	Authors	Cancer Entity	Treatment	Main Factor for Altered Response to Chemotherapy	Effect on Chemotherapy	Effects	Ref.
2013	Viaud et al.	Melanoma, sarcoma	Cyclophosphamide	*Lactobacillus johnsonii, Enterococcus hirae*	Enhancing	*L. johnsonii* and *E. hirae* stimulate memory Th1 and “pathogenic” pTh17 cell immune responses	[104]
2013	Iida et al.	Lymphoma, CRC, melanoma	Oxaliplatin	Vancomycin, imipenem, neomycin	Inhibiting	Bacteria induce tumor-associated pro-inflammatory cells to produce reactive oxygen species, boosting platinum toxicity	[103]
2015	Lehouritis et al.	Several cancer cell lines	Several chemotherapeutics	*Escherichia coli, Listeria* *welshimeri*	Enhancing/Inhibiting	*E. coli* and *L. welshimeri* lead to impaired cytotoxicity of 10 commonly used chemotherapeutic agents while increasing the cytotoxic effect of 6 other drugs	[102]
2016	Pflug et al.	CLL, relapsed lymphoma	Cyclophosphamide, cisplatin	Vancomycin, teicoplanin, linezolid, daptomycin	Inhibiting	Patients receiving anti-Gram-positive antibiotics present significantly lower overall and median survival	[105]
2016	Daillère et al.	Melanoma, sarcoma	Cyclophosphamide	*E. hirae,* *Barnesiella intestinihominis*	Enhancing	*E. hirae* and *B. intestinihominis* shape tumor microenvironment reducing Treg cells and stimulating antitumor Th1 cell and cytotoxic T lymphocyte responses https://www.sciencedirect.com/topics/medicine-and-dentistry/t-cell-response	[106]
2017	Yu et al.	CRC	Oxaliplatin, 5-FU	*Fusobacterium nucleatum*	Inhibiting	*F. nucleatum* activates autophagy in CRC cells by stimulating TLR4 and MYD88 innate immune signaling preventing CRC cells from apoptosis	[107]
2017	Geller et al.	CRC/PDAC	Gemcitabine	*Gammaproteobacteria*	Inhibiting	Bacterial enzyme cytidine deaminase (mostly found in *Gammaproteobacteria*) metabolizes gemcitabine into its inactive form	[23]
2018	Yuan et al.	CRC	5-FU	Vancomycin, ampicillin, neomycin, metronidazole	Inhibiting	Mice treated with 5-FU and antibiotics show significantly higher tumor volume and lower α-diversity indicating potential disruption of the gut microbiome leading to impaired chemotherapy efficacy	[108]
2019	Zhang et al.	CRC	5-FU	*F. nucleatum*	Inhibiting	*F. nucleatum* induces upregulation of BIRC3, leading to inhibition of apoptosis and chemoresistance	[109]
2020	Nenclares et al.	Head and neck cancer	Platinum-based chemoradiation	Penicillin and derivatives, macrolides, quinolones	Inhibiting	Patients receiving broad-spectrum antibiotics present with significantly lower overall and disease-specific survival	[110]
2020	Roberti et al.	CRC	Oxaliplatin-based chemotherapy	*Bacteroides fragilis*, members of the family *Erysipelotrichaceae*	Enhancing	Intestinal commensals- and oxaliplatin-induced epithelial cell death leads to accumulation of Tfh cells promoting antitumor effector/memory CD8^+^ T cells	[111]
2021	Zhao et al.	Lung cancer	Platinum-based chemotherapy	*Streptococcus mutans* (R)*, Enterococcus casseliflavus* (R), *Leuconostoc lactis* (NR)*, Eubacterium siraeum* (NR)	Enriched in R/ NR group	Stool microbial composition associated with clinical outcomes of cancer patients: species, enriched in responder (R) and non-responder (NR) groups, serving as potential biomarker for chemotherapy response	[112]
2021	He et al.	CRC	Oxaliplatin	Butyrate	Enhancing	Butyrate promotes ID2-dependent IL-12 pathway enhancing antitumor CD8^+^ T cell responses	[113]
2022	Guenther et al.	PDAC	Gemcitabine	Gram-negative bacteria	Inhibiting	Lipopolysaccharide (LPS) as a surrogate for bacterial colonization of Gram-negative bacteria act as negative predictor for adjuvant gemcitabine efficacy: LPS confers worse disease-free and overall survival	[91]
2022	Panebianco et al.	PDAC	Gemcitabine	Butyrate	Enhancing	Butyrate enhances gemcitabine effectiveness by inducing apoptosis and reduces vessel associated stromal markers indicating reduced stromatogenesis	[114]
2022	Kesh et al.	PDAC	Gemcitabine, Paclitaxel	Queuosine (Q)	Inhibiting	Q impairs chemotherapy effectiveness by upregulating PRDX1 expression protecting against chemotherapy induced oxidative stress	[115]

T helper 1 cell (Th1 cell), pathogenic T helper 17 cell (pTh17 cell), colorectal cancer (CRC), chronic lymphocytic leukemia (CLL), regulatory T cell (Treg cell), 5-fluorouracil (5-FU), toll-like receptor 4 (TLR4), myeloid differentiation primary response 88 (MYD88), pancreatic ductal adenocarcinoma (PDAC), baculoviral IAP repeat-containing 3 (BIRC3), T follicular helper cell (Tfh cell), responder (R), non-responder (NR), inhibitor of DNA binding 2 (ID2), interleukin-12 (IL-12), lipopolysaccharide (LPS), queuosine (Q), Peroxiredoxin-1 (PRDX1).

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
