# Peer review of "The Microbiome in PDAC—Vantage Point for Future Therapies?"

_cancers, 2022, doi:10.3390/cancers14235974_

Round 1

Reviewer 1 Report

The Authors presents a well-written and structured review titled « The microbiome in PDAC – vantage point for future therapies ? ». Several reviews on the field have been published over the last few years, therefore some overlap with previous reviews is inevitable. I appreciated the fact that contamination issues were addressed and that this review is aimed at both scientitsts and clinicians. The two figures provided are clear and well structured. 

Main comments :

-     The discussion about contamination issues and how to perform in the future quality trials regarding the PDAC microbiota is a strenght for this review and should be further emphazized and highlighted. This should also be mentionned in the abstract.

-     The chapter « Blood-derived microbial signatures as PDAC biomarker » should be expanded and new interesting techniques like the study of bacterial extracellular vesicles in PDAC should be mentioned.

-      The tables 1 -2 - 3 are mentioned in the text but are missing in the manuscript provided for peer review.

-      I suggest to better highlight and further discuss the perspective of microbiome-related treatments options . Also, in order to give a better perspective to the readers, interesting ongoing trials in the field should be mentionned. 

Specific comments :

-       A definition of microbiome/microbiota should be added in the introduction.

-      Introduction : it could be interesting to the readers to add that the microbiome was recently added in the new hallmarks of cancer (Hanahan et al. Cancer Discovery 2022).

-       Line number 40-41 : « high intrinsic resistance to chemotherapy » should be changed in « high-intrinsic resistance to systemic therapies and radiotherapy »

-      Lines 19 and 85 : « diseased pancreatic tissue » should be changed in « tumoral pancreatic tissue » as it is the only pancreatic disease discussed in the related chapter.

-     Lines 169-172 : « In other words, microbes were considered contaminants whose abundance did not increase with increasing read number following the assumption that contaminants should not correlate with total read numbers but affect all samples equally. » Long and unclear sentence.

Author Response

Reviewer 1

Comments and Suggestions for Authors

The Authors presents a well-written and structured review titled « The microbiome in PDAC – vantage point for future therapies ? ». Several reviews on the field have been published over the last few years, therefore some overlap with previous reviews is inevitable. I appreciated the fact that contamination issues were addressed and that this review is aimed at both scientitsts and clinicians. The two figures provided are clear and well structured. 

Response: First, we would like to thank the reviewer for the detailed analysis of our manuscript and the constructive comments that we appreciate very much.

Main comments :

-     The discussion about contamination issues and how to perform in the future quality trials regarding the PDAC microbiota is a strenght for this review and should be further emphazized and highlighted. This should also be mentionned in the abstract.

Response: We appreciate that the reviewer supports the chapter about decontamination approaches as we also believe that this is one of the major pitfalls. We implemented emphasized statements among others in abstract, introduction and conclusion underlining the contamination problem. 

-     The chapter « Blood-derived microbial signatures as PDAC biomarker » should be expanded and new interesting techniques like the study of bacterial extracellular vesicles in PDAC should be mentioned.

Response: We thank the reviewer for the constructive suggestion. Bacterial extracellular vesicles represent a very interesting field of research with high potential of further diagnostic and therapeutic approaches. To this end, we expanded our manuscript giving an overview about applications of bEVs in cancer in general and in PDAC.  

-      The tables 1 -2 - 3 are mentioned in the text but are missing in the manuscript provided for peer review.

Response: We apologize for the incomplete content that was an uploading issue. All tables are attached in the current uploaded file.

-      I suggest to better highlight and further discuss the perspective of microbiome-related treatments options . Also, in order to give a better perspective to the readers, interesting ongoing trials in the field should be mentionned. 

Response: Here again we are in line with the reviewer’s suggestion that future perspectives needed to be mentioned as well. We added a section giving an overview about current clinical trials registered at ClinicalTrial.gov.

Specific comments :

-      A definition of microbiome/microbiota should be added in the introduction.

Response: A definition and a brief description of the microbiome was added in the introduction.

-      Introduction : it could be interesting to the readers to add that the microbiome was recently added in the new hallmarks of cancer (Hanahan et al. Cancer Discovery 2022).

Response: We thank the reviewer for the advice and added the recent publication by Hanahan et al. in the introduction.

-       Line number 40-41 : « high intrinsic resistance to chemotherapy » should be changed in « high-intrinsic resistance to systemic therapies and radiotherapy »

Response: This passage was changed according to the reviewer’s suggestion.

-      Lines 19 and 85 : « diseased pancreatic tissue » should be changed in « tumoral pancreatic tissue » as it is the only pancreatic disease discussed in the related chapter.

Response:Diseased” was changed to “tumoral” according to the reviewer’s advice.

-     Lines 169-172 : « In other words, microbes were considered contaminants whose abundance did not increase with increasing read number following the assumption that contaminants should not correlate with total read numbers but affect all samples equally. » Long and unclear sentence.

Response: We changed this part and hope that it is clearer and understandable.

Reviewer 2 Report

This article provides an important synthesis of the current landscape with regards to PDAC, its investigation to date, methodological pitfalls to current investigative techniques, and future therapeutic perspectives. It is significant, especially in the critique it provides to current methodologies utilised to interrogate the microbiome in PDAC. Whilst these are strong positives to the article, it suffers from significant pitfalls. 

The first of these is the lack of tables. There are three tables referred to in the text. Unfortunately, none of these are provided in the text or as additional downloads/supplementary figures. These need to be added or reference to them needs to be removed from the text. 

Additionally, it suffers from poor referencing. There are 111 references listed in the references section but only 82 listed in the body of the text. The authors need to review their text and ensure it is appropriately referenced, with the text fully and appropriately references.

There are certain crucial references missing. I leave it to the authors discretion, but I strongly believe that Hanahan’s 2022 update to the Hallmarks of cancer framework should be included. 

Douglas Hanahan; Hallmarks of Cancer: New Dimensions. Cancer Discov 1 January 2022; 12 (1): 31–46. https://doi.org/10.1158/2159-8290.CD-21-1059

Furthermore, the text suffers from severe grammatical issues. As well as structural issues. The authors may wish to consult an English writing/checking service. 

Lines 58 – 79 need significant and careful review of the written English.

Th point made in lines 93 – 95 needs to be carefully examined and reviewed by the authors. This currently reads as:

A competing theory is a microbial colonialization of distant organs via the blood stream [20,21]. 

This is a bold statement and insinuates a constant subclinical level of bacteraemia in PDAC or bacteraemia as a means of microbial colonisation. The references provided refer to the evaluation of the 16s rRNA in the peripheral blood of patients with liver cirrhosis and the other looks at circulating microbial DNA in the blood of patients with pneumonia. There is no further expansion of this statement with regards to PDAC, with the rest of the paragraph referring to migration to PDAC following oral gavage. If the authors stand by their statement, they need to support this with strong evidence referring to PDAC. Within the bounds of my understanding, the presence of cell free DNA or RNA does not necessarily denote circulating bacteria and DNA or RNA can circulate from cell turnover, cancer, or infection. Indeed, cell free DNA can be utilised in screening for disease in foetuses (testing the mother’s blood for foetal DNA and utilising this for genetic testing) or in cancer research as an avenue for cancer diagnosis, disease progression monitoring and drug resistance assessment (occasionally referred to as liquid biopsy). Thus, extrapolating from the presence of DNA or RNA in the blood to the conclusion that there is haematogenous spread is a significant leap without supporting mechanistic evidence specifically in PDAC (or cancers more broadly at a minimum). It is recommended that the authors abandon this sentence unless they can strongly support this, in which case this has paradigm shifting implications. 

There are grammatical deficiencies in line 107. I suggest:

PDAC material [9]. Contrary, Thomas et al. did not find any significant differences be- 

Should read:

PDAC material [9]. Contrary to this Thomas et al. did not find any significant differences be- 

Table 1 is referred to but not provided.

Lines 97 to 199, although in need of some grammatical refinement, make a significant contribution to the methodological critiques to the microbiome in PDAC. This content was enjoyable to read, despite the need for grammatical adjustment and contains some of the strongest material in the paper. 

The authors refer to the microbiome in activating oncogenic signalling but do not seem to make any clear reference to whether any of this mechanistic work has been done specifically in PDAC. If no data exist on this in PDAC specifically, the authors may wish to consider calling for the need for this to be investigated to raise this to the wider field. 

There are grammatical adjustments needed in lines 223 to 224, 244 – 251. 

Line 241 – 242 is unsubstantiated in this statement 

            These are mainly accompanied by chronic inflammation. 

No reference is provided, and the mechanism described refers to commensal MAMPs, with no reference to chronic inflammation in the rest of the pancreas. This section has potential if appropriately researched and expanded, but at present lacks in the potential positive contribution it could make to the paper. 

The paragraph beginning from line 252 to 265 may benefit from some restructuring. 

Lines 267 – 280 requires grammatical correction. Additionally, the term ‘carcinogenesis’ is incorrectly used, as the Pushalker et al (as the authors refer to in the paragraph above) found reduced tumour burden (which can be taken as growth), not a reduction in tumorigenesis, which suggests cancers forming de novo.

Line 280 is unclear in what was genetically deleted. This should be clarified. 

Lines 281 to 294 could do with some grammatical adjustment. Additionally, it seems to expand on lines 244 – 250. The authors should expand on the earlier section to make it distinct as it has promise, although some of the ideas above may be best combined with this later paragraph. 

There are grammatical adjustments that need to be made in the paragraph from 297 – 304.

Table 2 is not provided. 

Grammatical adjustment needed in line 312. 

Line 339 needs a reference. This section can be expanded if the authors wish (see comments above about cell free DNA/RNA and liquid biopsy).

Table 3 is not provided. 

Grammatical corrections needed in section from lines 377 to 382.

384 – 388 needs grammatical adjustment.

Expanding on the mechanistic aspects of the following would benefit the paper.  

Kesh and colleagues most recently found that FMT from PDAC-implanted control mice to PDAC-implanted obese mice rendered the tumors of the latter more sensitive towards chemotherapy [80]. S-adenosyl methionine-producing bacteria were enriched in the feces of mice who received the control diet, whereas queuosine-producing bacteria were elevated in the high fat diet-fed mice. This emphasizes why dietary modifications may be beneficial to enhance anticancer therapy. 

A large amount of the section entitled “the microbiome as a potential biomarker for PDAC prognosis” seems to lack focus. Some of the contents here may be better placed in the section on “Microbiome modulation approaches” or “Microbial impact on anti-tumour therapies.” 

Lines 436-441 are deserving of expanding upon as it is significant and is deserving of a paragraph. Additionally, the ideas are distinct from the preceding sentence so would be grammatically indicated. 

Overall there are promising elements to this review and I would like to see the authors succeed in publishing their work. It is clear they have taken the time to generate an important and relevant synthesis of the field, with important insights. Nevertheless, there is a significant amount of work yet to be done to bring this manuscript up to publication standard. 

Author Response

Reviewer 2

Comments and Suggestions for Authors

This article provides an important synthesis of the current landscape with regards to PDAC, its investigation to date, methodological pitfalls to current investigative techniques, and future therapeutic perspectives. It is significant, especially in the critique it provides to current methodologies utilised to interrogate the microbiome in PDAC. Whilst these are strong positives to the article, it suffers from significant pitfalls. 

Response:  We like to thank the reviewer for the detailed analysis of our manuscript and the constructive comments that are very helpful to establish a more comprehensive and clearer manuscript.

The first of these is the lack of tables. There are three tables referred to in the text. Unfortunately, none of these are provided in the text or as additional downloads/supplementary figures. These need to be added or reference to them needs to be removed from the text. 

Response:  We apologize for the incomplete content that was an uploading issue. All tables are attached in the current uploaded file. These tables are an integral part of the manuscript providing a tabular summary for questions the readers might have on this topic.

Additionally, it suffers from poor referencing. There are 111 references listed in the references section but only 82 listed in the body of the text. The authors need to review their text and ensure it is appropriately referenced, with the text fully and appropriately references.

Response:  Here again we apologize for this shortcoming. The reason for this discrepancy is the missing tables. They contain the remaining references which were summarized but not directly mentioned in the main script. The current uploaded version includes the missing tables and thus all references are also found in the manuscript.   

There are certain crucial references missing. I leave it to the authors discretion, but I strongly believe that Hanahan’s 2022 update to the Hallmarks of cancer framework should be included. 

Douglas Hanahan; Hallmarks of Cancer: New Dimensions. Cancer Discov 1 January 2022; 12 (1): 31–46. https://doi.org/10.1158/2159-8290.CD-21-1059

Response: We thank the reviewer for the advice and added the recent publication by Hanahan et al. in the introduction.

Furthermore, the text suffers from severe grammatical issues. As well as structural issues. The authors may wish to consult an English writing/checking service. 

Lines 58 – 79 need significant and careful review of the written English.

Response: We carefully revised this passage and hope it improves significantly to meet the reviewer’s and the journal’s expectations.

Th point made in lines 93 – 95 needs to be carefully examined and reviewed by the authors. This currently reads as:

A competing theory is a microbial colonialization of distant organs via the blood stream [20,21]. 

This is a bold statement and insinuates a constant subclinical level of bacteraemia in PDAC or bacteraemia as a means of microbial colonisation. The references provided refer to the evaluation of the 16s rRNA in the peripheral blood of patients with liver cirrhosis and the other looks at circulating microbial DNA in the blood of patients with pneumonia. There is no further expansion of this statement with regards to PDAC, with the rest of the paragraph referring to migration to PDAC following oral gavage. If the authors stand by their statement, they need to support this with strong evidence referring to PDAC. Within the bounds of my understanding, the presence of cell free DNA or RNA does not necessarily denote circulating bacteria and DNA or RNA can circulate from cell turnover, cancer, or infection. Indeed, cell free DNA can be utilised in screening for disease in foetuses (testing the mother’s blood for foetal DNA and utilising this for genetic testing) or in cancer research as an avenue for cancer diagnosis, disease progression monitoring and drug resistance assessment (occasionally referred to as liquid biopsy). Thus, extrapolating from the presence of DNA or RNA in the blood to the conclusion that there is haematogenous spread is a significant leap without supporting mechanistic evidence specifically in PDAC (or cancers more broadly at a minimum). It is recommended that the authors abandon this sentence unless they can strongly support this, in which case this has paradigm shifting implications. 

Response: We are in line with the reviewer that our statement was highly theoretical and that the cited references do not strongly support this hypothesis. We deleted this sentence to avoid confusions as we have not found further supporting literature for our statement.

There are grammatical deficiencies in line 107. I suggest:

PDAC material [9]. Contrary, Thomas et al. did not find any significant differences be- 

Should read:

PDAC material [9]. Contrary to this Thomas et al. did not find any significant differences be- 

Response:  We changed this sentence according to the reviewer’s suggestion.

Table 1 is referred to but not provided.

Response:  Please refer to the response above.

Lines 97 to 199, although in need of some grammatical refinement, make a significant contribution to the methodological critiques to the microbiome in PDAC. This content was enjoyable to read, despite the need for grammatical adjustment and contains some of the strongest material in the paper. 

The authors refer to the microbiome in activating oncogenic signalling but do not seem to make any clear reference to whether any of this mechanistic work has been done specifically in PDAC. If no data exist on this in PDAC specifically, the authors may wish to consider calling for the need for this to be investigated to raise this to the wider field. 

There are grammatical adjustments needed in lines 223 to 224, 244 – 251. 

Response: We carefully revised these passages. 

Line 241 – 242 is unsubstantiated in this statement 

            These are mainly accompanied by chronic inflammation. 

No reference is provided, and the mechanism described refers to commensal MAMPs, with no reference to chronic inflammation in the rest of the pancreas. This section has potential if appropriately researched and expanded, but at present lacks in the potential positive contribution it could make to the paper. 

Response: Again, the reviewer pointed out an important issue. All studies suggesting the link between periodontitis and PDAC have several limitations. To this end, we agree with the reviewer that statements like above are not appropriate. We added further details from the cited references and discuss the results profoundly.

The paragraph beginning from line 252 to 265 may benefit from some restructuring. 

Response: We carefully revised this paragraph. 

Lines 267 – 280 requires grammatical correction. Additionally, the term ‘carcinogenesis’ is incorrectly used, as the Pushalker et al (as the authors refer to in the paragraph above) found reduced tumour burden (which can be taken as growth), not a reduction in tumorigenesis, which suggests cancers forming de novo.

Response: We conducted changes as recommended.

Line 280 is unclear in what was genetically deleted. This should be clarified. 

Response: This paragraph was adjusted and clarified.

Lines 281 to 294 could do with some grammatical adjustment. Additionally, it seems to expand on lines 244 – 250. The authors should expand on the earlier section to make it distinct as it has promise, although some of the ideas above may be best combined with this later paragraph. 

Response: We totally agree that mentioning different LPS studies in different chapters are redundant and thus adjusted both paragraphs.

There are grammatical adjustments that need to be made in the paragraph from 297 – 304.

Response: We carefully revised this paragraph. 

Table 2 is not provided. 

Response: Please refer to the response above.

Grammatical adjustment needed in line 312. 

Response: We carefully revised this paragraph. 

Line 339 needs a reference. This section can be expanded if the authors wish (see comments above about cell free DNA/RNA and liquid biopsy).

Response: We expanded the blood-derived biomarkers section and provided more details about bacterial extracellular vesicles (bEVs).

Table 3 is not provided. 

Response: Please refer to the response above.

Grammatical corrections needed in section from lines 377 to 382.

384 – 388 needs grammatical adjustment.

Response: We carefully revised both paragraphs. 

Expanding on the mechanistic aspects of the following would benefit the paper.  

Kesh and colleagues most recently found that FMT from PDAC-implanted control mice to PDAC-implanted obese mice rendered the tumors of the latter more sensitive towards chemotherapy [80]. S-adenosyl methionine-producing bacteria were enriched in the feces of mice who received the control diet, whereas queuosine-producing bacteria were elevated in the high fat diet-fed mice. This emphasizes why dietary modifications may be beneficial to enhance anticancer therapy. 

Response: More details about the mechanisms were added from the well-designed study by Kesh et al.

A large amount of the section entitled “the microbiome as a potential biomarker for PDAC prognosis” seems to lack focus. Some of the contents here may be better placed in the section on “Microbiome modulation approaches” or “Microbial impact on anti-tumour therapies.” 

Response: We agree that the study by Guo et al. describe rather the microbial impact on anti-tumor therapies and shifted this passage accordingly.

Lines 436-441 are deserving of expanding upon as it is significant and is deserving of a paragraph. Additionally, the ideas are distinct from the preceding sentence so would be grammatically indicated. 

Response: We expand more details from the study published by Chakaldar et al.

Reviewer 3 Report

The title of the paper suggests that future therapies will be the overall focus, although there is very little argument made for what therapies may be explored in the future or which ones the authors feel have the most promise.

This paper is a review of the topic of the microbiome in PDAC, but there is no description of what the microbiome is or why it is relevant. There is no mention of what constitutes the microbiome, no mention of the intestinal microbiome as the largest in the human body and the one that is being investigated more than the others. The authors do not highlight that there is a pool of evidence suggesting that it is clinically targetable and that there is evidence in PDAC that manipulating the gut microbiome can alter the tumour microbiome in murine models.

There are clinical trials that have been published showing that manipulating the intestinal microbiome is being tried in the clinic. The authors mention immune checkpoint inhibitors (ICI), but do not highlight how they have revolutionized treatment of many cancers, no mention of clinical trials that have not shown any benefit in PDAC, or how clinical trials in melanoma published in the last 2 years have shown that FMT can sensitize resistant tumours to immune checkpoint inhibitors. The field of PDAC research is very focused on using FMT to make PDAC sensitive to ICI and hopefully improve the prognosis of a bad cancer similar to what has been done for melanoma and lung cancer.

The authors suggest this is a paper focused on future treatments and they mention clinicians and translational scientists several times. When discussing PDAC, there is no mention to the readers that this is a poor prognosis cancer AND current therapies are both limited in number and benefit. There is no mention that most are diagnosed at the unresectable stage and systemic therapy is the only treatment option. They do mention several reasons why they are hard to treat to include therapy-resistance as well as high potential for metastases. These are not incorrect points, but they miss what PDAC really looks like in the clinic - not sure if the physician authors treat PDAC? It just isn't clear that they are experienced in treating PDAC and understand the urgency and need for better treatments. PDAC is a difficult cancer to treat because it is essentially a systemic disease before it is found in the majority of patients and our treatments are very modestly beneficial. PDAC does not need to be metastatic at the time of diagnosis to be advanced and imminently life-threatening. Unresectable PDAC is nearly as bad, although patients may survive a short time longer. There is no mention that even those patients who undergo resection usually have it recur. I just don't get the sense that the motivation for new therapies is conveyed to the readers.

There is some space dedicated to discussing the tumour's resistance to gemcitabine due cytidine deaminase produced by certain microbes in the tumour microenvironment, but no context of why this matters. There are only 5 drugs with activity and clinical evidence in PDAC. Two of the 3 regimens include gemcitabine, and many patients receive gemcitabine-based therapy because they are not fit enough for FOLFIRINOX. It isnt enough to say that FOLFIRINOX is the tougher regimen, readers need some context. There is a great deal of information/data in the literature to discuss these details in more depth. The inclusion of several lines on the metabolism of oxaliplatin is not helpful. Oxaliplatin is not the backbone of the regimen as the authors state - just seeming like the authors do not have experience in treating PDAC. This would be fine but the title of the paper suggests it is about the treatment of PDAC. It is important for readers to understand what the state of treatment is like now. Many readers may not know much about PDAC and some context is important.

There is a brief introduction to oncomicrobes, but this is not relevant for this type of article. Readers who may not know much about oncomicrobes or the microbiome might think that they are similar. Pathogenic microbes are very different from the commensal organisms that perform several beneficial functions for the human body - many functions that are helpful and that our bodies cannot do for themselves. It is not made clear that not all 'bugs' are bad. If this is a review, it is intended to educate.

Several concepts are mentioned, but then little evidence in PDAC is presented, often relying on what has been identified in cancers such as colorectal cancer (CRC). There is a lot of research in PDAC that is available to draw on. But when there is not, the authors do not clearly explain how some evidence in CRC is relatable to PDAC

Towards the end, there is a discussion about long term survivors (LTS) in PDAC as being prognostic. Identifying the rare LTS is not useful in the clinic. The data was presented to help identify what makes LTS patients do better than the vast majority of patients with PDAC. The LTS patients received the standard of care treatment, so knowing who they are at the time of diagnosis would not change management, and would suggest that for those patients we do not need better therapeutics. The paper that described LTS also provided excellent evidence for using the gut microbiome to manipulate the pancreatic microbiome in murine models, but this important data was not mentioned in the discussion about modifying the tumour microbiome in PDAC.

The authors spend some time discussing the potential for using the microbiome as a screening tool to identify patients with PDAC. They review some of the observational evidence but make no mention of whether this is practical or not at the population level. They do not suggest who could be screened (ie. high risk, everyone?), or discuss how this information could be used - is early detection of PDAC likely to change the outcome? It is an aggressive tumour with poor treatment options. Is it likely to help find patients with PDAC early enough to allow for a curative surgery? There is only a description of retrospective evidence but no commentary on how it may benefit patients. 

Finally, the title suggests therapeutics for PDAC is the focus, but there is a significant amount of the text dedicated to methods and techniques to properly evaluate the microbiome and how to avoid contamination. The authors seem to want to write a paper with two different topics put together. The addition of such a large amount of lab technique in a review of therapies does not fit. There can be a separate paper reviewing the methods for accurate identification of the taxa and species in the microbiome, and a separate paper reviewing therapeutics. All but the last line in the abstract leads the reader to expect the paper to be focused on the clinical and translational science aspect of the role of the microbiome in PDAC - not lab technique.

My recommendation to the readers is to eliminate the discussion on how to isolate the microbiota. Focus on future therapeutics or technology, but not both together. The evidence that is presented needs to have context and the authors need to discuss why they think this is or could be important in PDAC. A brief description of the human microbiome should be included in a review, it does not need to be extensive, but the reader needs to have some understanding of what it is and why it is important in PDAC. The review needs to highlight the current state of PDAC treatment to help readers understand how bad it is clinically, for real patients. This would then lead into a discussion of why it is so hard to treat. If the authors do not treat PDAC themselves, the paper may benefit from the perspective of physicians who do treat it.

Author Response

Reviewer 3

Response: We also like to thank the third reviewer for the thorough reading of our manuscript and comments. We are willing to discuss and consider the following constructive points for further adjustments of the manuscript.

The title of the paper suggests that future therapies will be the overall focus, although there is very little argument made for what therapies may be explored in the future or which ones the authors feel have the most promise.

Response: We strongly believe that the microbiome contributes to tumorigenesis, response to chemotherapy and prognosis of this dismal disease. However, during the literature research we came across of huge discrepancies regarding microbial compositions between studies and their conclusions. Thus, we have to summarize the question raised in the title that the microbiome is not ready for being exploited as a diagnostic or therapeutic tool. But we are in line with the reviewer that this statement has to be more emphasized. To this end, adjustments in the abstract and conclusion were done.

This paper is a review of the topic of the microbiome in PDAC, but there is no description of what the microbiome is or why it is relevant. There is no mention of what constitutes the microbiome, no mention of the intestinal microbiome as the largest in the human body and the one that is being investigated more than the others. The authors do not highlight that there is a pool of evidence suggesting that it is clinically targetable and that there is evidence in PDAC that manipulating the gut microbiome can alter the tumour microbiome in murine models.

Response: A definition of microbiome and additional details are now provided in the introduction. In our section “Microbiome modulation approaches”, we summarize the current literature circling around the clinical targeting of the microbiome and how to potentially alter it in a favourable way. If we lack important publications, we highly appreciate the reviewer’s suggestions.

There are clinical trials that have been published showing that manipulating the intestinal microbiome is being tried in the clinic. The authors mention immune checkpoint inhibitors (ICI), but do not highlight how they have revolutionized treatment of many cancers, no mention of clinical trials that have not shown any benefit in PDAC, or how clinical trials in melanoma published in the last 2 years have shown that FMT can sensitize resistant tumours to immune checkpoint inhibitors. The field of PDAC research is very focused on using FMT to make PDAC sensitive to ICI and hopefully improve the prognosis of a bad cancer similar to what has been done for melanoma and lung cancer.

Response: We thank the reviewer for this constructive suggestion. We added the clinical phase 1 trials which explore the effectiveness of FMT in primary ICI non-responders. However, even a thorough search at trial.gov has not been fruitful in terms of ongoing trials of FMT in PDAC upon ICI treatment. In case the reviewer knows about certain publications/ ongoing trials, we would highly appreciate it if this information could be shared.

The authors suggest this is a paper focused on future treatments and they mention clinicians and translational scientists several times. When discussing PDAC, there is no mention to the readers that this is a poor prognosis cancer AND current therapies are both limited in number and benefit. There is no mention that most are diagnosed at the unresectable stage and systemic therapy is the only treatment option. They do mention several reasons why they are hard to treat to include therapy-resistance as well as high potential for metastases. These are not incorrect points, but they miss what PDAC really looks like in the clinic - not sure if the physician authors treat PDAC? It just isn't clear that they are experienced in treating PDAC and understand the urgency and need for better treatments. PDAC is a difficult cancer to treat because it is essentially a systemic disease before it is found in the majority of patients and our treatments are very modestly beneficial. PDAC does not need to be metastatic at the time of diagnosis to be advanced and imminently life-threatening. Unresectable PDAC is nearly as bad, although patients may survive a short time longer. There is no mention that even those patients who undergo resection usually have it recur. I just don't get the sense that the motivation for new therapies is conveyed to the readers.

Response: In the introduction we already mentioned the poor outcome of PDAC and listed and explained different reasons. However, according to reviewer’s recommendations we added more epidemiological information to underline this devasting prognosis.

There is some space dedicated to discussing the tumour's resistance to gemcitabine due cytidine deaminase produced by certain microbes in the tumour microenvironment, but no context of why this matters. There are only 5 drugs with activity and clinical evidence in PDAC. Two of the 3 regimens include gemcitabine, and many patients receive gemcitabine-based therapy because they are not fit enough for FOLFIRINOX. It isnt enough to say that FOLFIRINOX is the tougher regimen, readers need some context. There is a great deal of information/data in the literature to discuss these details in more depth. The inclusion of several lines on the metabolism of oxaliplatin is not helpful. Oxaliplatin is not the backbone of the regimen as the authors state - just seeming like the authors do not have experience in treating PDAC. This would be fine but the title of the paper suggests it is about the treatment of PDAC. It is important for readers to understand what the state of treatment is like now. Many readers may not know much about PDAC and some context is important.

Response: We thank the reviewer for the advice to emphasize the clinical importance of microbiome-driven resistance against Gemcitabine. We added statements accordingly. An overview about the different chemotherapy regimen is provided in the introduction as mentioned in the response above. Furthermore, we also adjusted the passage with Oxaliplatin.

There is a brief introduction to oncomicrobes, but this is not relevant for this type of article. Readers who may not know much about oncomicrobes or the microbiome might think that they are similar. Pathogenic microbes are very different from the commensal organisms that perform several beneficial functions for the human body - many functions that are helpful and that our bodies cannot do for themselves. It is not made clear that not all 'bugs' are bad. If this is a review, it is intended to educate.

Response: The knowledge about an intratumoral microbiome is novel. Most readers who are not familiar with this topic might have traditional oncomicrobes in mind when reading about cancer and microbes. To point out the sharp contrast to these well-defined oncomicrobes, we decided to give a brief introduction to these traditional pathogens. The fact that the host microbiome can be also beneficial is emphasized in several parts. Please refer to sections such as “Microbial impact on anti-tumour therapies” or “The microbiome as potential biomarker for PDAC prognosis”, for instance.

Several concepts are mentioned, but then little evidence in PDAC is presented, often relying on what has been identified in cancers such as colorectal cancer (CRC). There is a lot of research in PDAC that is available to draw on. But when there is not, the authors do not clearly explain how some evidence in CRC is relatable to PDAC

Response: To the best of our knowledge we summarized the current literature of carcinogenesis concepts linked to the microbiome. If the experiment or the clinical trial was not performed with PDAC patients/ models, we provided information which tumour (-model) was analysed and refer to potential crosslinks in PDAC. Please refer to figure 2. If there is a lack of specific references or relatable links, we highly appreciate precise suggestions from the reviewer.   

Towards the end, there is a discussion about long term survivors (LTS) in PDAC as being prognostic. Identifying the rare LTS is not useful in the clinic. The data was presented to help identify what makes LTS patients do better than the vast majority of patients with PDAC. The LTS patients received the standard of care treatment, so knowing who they are at the time of diagnosis would not change management, and would suggest that for those patients we do not need better therapeutics. The paper that described LTS also provided excellent evidence for using the gut microbiome to manipulate the pancreatic microbiome in murine models, but this important data was not mentioned in the discussion about modifying the tumour microbiome in PDAC.

Response: We agree with the reviewer that Riquelme et al. not only provides evidence for the intratumoral microbiome as potential biomarker for PDAC prognosis but also proved that FMT is a potential therapeutic option to change the intratumoral microbiome towards a more favourable composition. Thus, we shifted this part to the section “Microbiome modulation approaches”.

The authors spend some time discussing the potential for using the microbiome as a screening tool to identify patients with PDAC. They review some of the observational evidence but make no mention of whether this is practical or not at the population level. They do not suggest who could be screened (ie. high risk, everyone?), or discuss how this information could be used - is early detection of PDAC likely to change the outcome? It is an aggressive tumour with poor treatment options. Is it likely to help find patients with PDAC early enough to allow for a curative surgery? There is only a description of retrospective evidence but no commentary on how it may benefit patients. 

Response: The reviewer indicates an important issue. We added a discussion about the practical translation of the microbiome as a non-invasive biomarker for PDAC.

Finally, the title suggests therapeutics for PDAC is the focus, but there is a significant amount of the text dedicated to methods and techniques to properly evaluate the microbiome and how to avoid contamination. The authors seem to want to write a paper with two different topics put together. The addition of such a large amount of lab technique in a review of therapies does not fit. There can be a separate paper reviewing the methods for accurate identification of the taxa and species in the microbiome, and a separate paper reviewing therapeutics. All but the last line in the abstract leads the reader to expect the paper to be focused on the clinical and translational science aspect of the role of the microbiome in PDAC - not lab technique.

Response: From our point of view, it is crucial to include technical aspects, such as decontamination, in a review dealing with therapeutic approaches. Even clinicians have to be aware of the pitfalls that are inherent to microbiome research. We like to illustrate this thesis with an example beyond those provided in the manuscript. The hallmark study by Riquelme et al. could clearly show that the intratumoral microbiomes of LTS and STS are totally distinguishable. They also proved causality with a FMT mouse experiment which was also remarked by the reviewer. However, genera which display the most significant difference between LTS and STS are:

  • Streptomyces
  • Pseudoxanthomonas
  • Saccharopolyspora

All three taxa are rarely seen in clinical specimen, such as blood or urine cultures. They are also not the common intestinal commensals published in gut microbiome studies. A brief look into PubMed literature revealed that all three genera are rather environmental bacteria found in mud, marine sponge and/ or soil. It might be possible that these untypical taxa also occupy certain human body sites. However, for a definitive proof of their existence a well-designed methodology is crucial. A glimpse into the method part of Riquelme et al. tells the reader that only 4 negative controls were used for all samples. This raises concerns that the key taxa of this high impact study are not really bacteria from the tumour at all.

If we wish to develop new clinical therapeutic approaches to tackle the dismal prognosis of PDAC, we have to know the real microbial composition in the tumour before starting to think about a modulation of the microbiome, especially for more precisely targeted treatments like the supplementation of pro-, pre- or postbiotics. We hope that this small example convinced the reviewer about the necessity of presenting both laboratory methodology and clinical perspectives in one review. To emphasize this statement we adjusted the abstract, conclusion and parts of the introduction.   

My recommendation to the readers is to eliminate the discussion on how to isolate the microbiota. Focus on future therapeutics or technology, but not both together. The evidence that is presented needs to have context and the authors need to discuss why they think this is or could be important in PDAC. A brief description of the human microbiome should be included in a review, it does not need to be extensive, but the reader needs to have some understanding of what it is and why it is important in PDAC. The review needs to highlight the current state of PDAC treatment to help readers understand how bad it is clinically, for real patients. This would then lead into a discussion of why it is so hard to treat. If the authors do not treat PDAC themselves, the paper may benefit from the perspective of physicians who do treat it.

Response: We thank the reviewer for the constructive comments and suggestion, particularly to sharpen the clinical aspects of our manuscript.

Above we discussed, adjusted, expanded and argued following issues:

  • The necessity of presenting both methodology and clinical aspects in this review
  • Providing a definition of the microbiome with a brief description
  • Providing current epidemiological basics underlying the dismal prognosis of PDAC

Round 2

Reviewer 1 Report

The author responses were statisfiying

Reviewer 2 Report

The authors have clearly taken on the comments made and have made adjustments to the paper reflecting these. I am supportive of publication.